# Optical Flow Estimation by Matching Time Surface with Event-Based Cameras

**DOI:** 10.3390/s21041150

**Published:** 2021-02-06

**Authors:** Jun Nagata, Yusuke Sekikawa, Yoshimitsu Aoki

**Affiliations:** 1Department of Electronics and Electrical Engineering, Faculty of Science and Technology, Keio University, 3-14-1, Hiyoshi, Kohoku-ku, Yokohama, Kanagawa 223-8522, Japan; aoki@elec.keio.ac.jp; 2Denso IT Laboratory, 2-15-1, Shibuya, Shibuya-ku, Tokyo 150-0002, Japan; ysekikawa@d-itlab.co.jp

**Keywords:** event-based camera, optical flow

## Abstract

In this work, we propose a novel method of estimating optical flow from event-based cameras by matching the time surface of events. The proposed loss function measures the timestamp consistency between the time surface formed by the latest timestamp of each pixel and the one that is slightly shifted in time. This makes it possible to estimate dense optical flows with high accuracy without restoring luminance or additional sensor information. In the experiment, we show that the gradient was more correct and the loss landscape was more stable than the variance loss in the motion compensation approach. In addition, we show that the optical flow can be estimated with high accuracy by optimization with L1 smoothness regularization using publicly available datasets.

## 1. Introduction

Event-based cameras are bio-inspired vision sensors that asynchronously output per-pixel brightness changes as the event stream instead of video frames [1]. The asynchronous nature of the event camera offers several advantages over traditional cameras, such as a high temporal resolution, a high dynamic range and low latency, which make event cameras more popular in many domains, such as mosaicing and tracking [2], 3D reconstruction [3], and high frame rate video generation [4]. Event-based cameras are also suitable for optical flow estimation since the precise timestamp at pixel-level intensity changes directly encode fine grain motion information.

However, the novel output of the camera, which is completely different from an image, provides new challenges in algorithm development. Several attempts have been made to apply a technique using spatiotemporal image derivatives and an assumption of brightness constancy [5,6] to event-based vision. Benosman et al. [7] and Brosch et al. [8] proposed that the spatial image derivative was approximated using the integration of events and applied to the optical flow constraint. Bardow et al. [9] estimated optical flow while simultaneously restoring image brightness from events only. It is preferable to utilize the precise timing information than to approximate the image gradient or restore the brightness using events for which absolute brightness information is lost.

Recently, motion compensation approaches that estimate the optical flow by searching for the trajectory that maximizes event alignment [10] have been developed. The best trajectory is obtained by maximizing the objective function that measures the edge sharpness of the image of warped events (IWE). However, Zhu et al. [11] pointed out that the variance loss leads to the overfitting of the networks by predicting the optical flows that make the events overlap in a line.

We also examined this phenomenon by visualizing the landscape and gradient of the loss function in various textures. We found that the gradient became larger in the direction pushing events to a line and the loss landscape was unstable around a line. Furthermore, the gradient is the spatial derivative of the surface formed by the warped event timestamp and may be incorrect where the surface is not continuous (e.g., around the latest event). As in the experiment in [10,12], if the optical flow parameter $R^{2}$ is common in the patch, the effect is small because the average is taken, but the effect is large in the pixel-level parameterization.

What we hope to achieve in this research is to estimate pixel-wise optical flow from events only, without additional sensor information, and without restoring the luminance. We focused on the time surface that retains the timestamp of the latest event at each pixel. The time surface is an aggregation of spatio-temporal events in the image form while keeping accurate timestamps that encodes motion information. To estimate the dense optical flow from the time surface, we propose Surface Matching Loss, which evaluates the timestamp consistency of the time surface between two times. By optimizing the proposed loss function with the smoothness regularization, dense optical flows can be estimated. An overview of the proposed method is shown in Figure 1.

In the experiment, the landscape of our loss function was shown to be gentle in any direction for any texture. In addition, we evaluated dense optical flow estimated with L1 smoothness regularization. The accuracy of the estimated optical flow was higher than with conventional methods, such as simultaneous estimation with image restoration [9] and the contrast maximization approach [10,12].

Our main contributions are summarized as follows:We propose the loss function measuring the timestamp consistency of the time surface for optical flow estimation using event-based cameras. This proposed loss function makes it possible to estimate dense optical flows without explicitly reconstructing image intensity or utilizing additional sensor information.Visualizing the loss landscape, we show that our loss is more stable regardless of the texture than the variance used in the motion compensation framework. Alongside this, we also show that the gradient is calculated in the correct direction in our method even around a line segment.We evaluate the dense optical flow estimated by optimization with L1 smoothness regularization. Our method recodes with higher accuracy compared with the conventional methods in the various scenes from the two publicly available datasets.

## 2. Related Work

There have been several works on estimating optical flows from event data in a novel data format. Benosman et al. [15] show that optical flow can be estimated directly from timing information by fitting a plane in x−y−t space. This approach succeeds in simple scenes with sharp edges, but fails in more complicated textures [16].

Benosman et al. [7] and Brosch et al. [8] applied the luminance gradient, which is approximated using the summation of events in the local spatiotemporal patch to the optical flow constraint [5,6]. Bardow et al. [9] proposed a method for simultaneous optical flow and intensity estimation from only event data. The proposed cost function is defined within a large sliding window containing a large number of images and optical flow parameters, and includes spatiotemporal smoothness regularization. However, the intermediation of the luminance information does not take full advantage of the precise timestamp of event-based cameras.

Several learning-based approaches [11,17,18] also exist. Zhu et al. [17] proposed a method for self-supervised learning for neural networks by photometric loss using grayscale images taken at the same time as events by Dynamic and Active-pixel Vision Sensor (DAVIS) [14]. It seems to be unable to learn in blacked-out or blurred scenes where brightness images can’t be taken. Ye et al. [18] also proposed a method for calculating photometric loss by warping event slices using optical flows generated from the depth and ego-motion output on the network.

The most recently studied method is the motion compensation framework [10,11,19,20,21,22], maximizing event alignment along point trajectories on the image plane. Event alignment is measured by various indices, such as the association probability of events in the EM algorithm [20], the number of overlapping events per pixel [21], the contrast of the IWE [10], and the average timestamp [11,22].

Zhu et al. [11] pointed out that the loss functions evaluating the image contrast [10] lead to the overfitting of the networks by estimating the optical flow that pushes the events into a line. Gallego et al. [12] investigated more than twenty loss functions to analyze event alignment. They reported that the accuracy of the average timestamp [11,22] was low, and the variance, the gradient and magnitude, and the Laplacian magnitudes are the best functions. Stoffregen et al. [23] analyzed the properties of contrast maximization reward functions and showed aperture uncertainty depending on data.

The gradient in the contrast maximization framework for optical flow estimation (refer to Equation (Equation 12) in Section 4.2) is calculated like the derivative of the time surface. In this calculation, the surface around the latest timestamp is spatially differentiated at discontinuous points, resulting in an incorrect gradient. In case of patch-based optical flow, the effect is small because the average is taken, but the effect is large in the pixel-level parameterization. In addition, we found that the gradient that makes events overlap in a line may be larger.

The aim of this study is to examine a method for estimating dense optical flow from events only, without additional sensor information and without recovering luminance. We focused on the common data representation of events, the time surface, which keeps the latest timestamp in the form of an image. To estimate the dense optical flow from the time surface, we propose a new Surface Matching Loss that measures the consistency of the timestamps on the time surface between two times. With this method, the gradient is more stable than that of the variance loss in any texture, and the loss landscape is gentle in any direction.

## 3. Methodology

### 3.1. Event Representation

Event-based cameras output events in response to changes in brightness for each pixel independently. An event $e_{k}=\left(x_{k},t_{k},p_{k}\right)$ is generated if the change in logarithmic brightness at a certain pixel reaches a pre-defined threshold. $x_{k}$, $t_{k}$, $p_{k}\in \left\{+,-\right\}$ represent pixel coordinates, the microsecond timestamp and polarity (i.e., sign of the brightness change), respectively.

### 3.2. Time Surface

The data representation commonly used in event-based vision is the surface of active events [15], also referred as time surface [24,25]. The time surface at a pixel $x={\left(x,y\right)}^{⊤}$ and polarity *p* is defined as
(1)$$S_{p}\left(x\right)\leftarrow t,\;\;\;\;p\in \left\{+,-\right\},$$
where *t* is the timestamp of the latest event with polarity *p* which is generated at pixel x. The time surface is a summary of the event-stream at each instant, which enables a scalable event-processing.

### 3.3. Surface Matching Loss

The event camera generates a trajectory of events in x−y−t space by the motion of an object. The time surface maintains the latest timestamp of the trajectory for each pixel. Consider the linear motion of an object over a very short time interval Δt within a rigid scene, represented by an arrow, as in Figure 2a. Using the displacement v(x) in time duration Δt at pixel x, the following equation is derived from the consistency of the timestamp:(2)$$S_{p}\left(x\right)=S_{p}\left(x+v\left(x\right)\right)-\Delta t.$$

The right-hand side can be considered to be the time surface shifted by Δt as shown in Figure 2b. Since the time surface consists of timestamps, it can be shifted in time at any time interval. The shifted time surface $S_{p}^{\prime }$ is defined as follows:(3)$$S_{p}^{\prime }\left(x\right):=S_{p}\left(x\right)-\Delta t.$$

The middle of Figure 1 shows a time surface $S_{p}$ and a shifted time surface $S_{p}^{\prime }$ at each polarity with actual event data [13]. Rewriting Equation (Equation 2) with $S_{p}^{\prime }\left(x\right)$ gives:(4)$$S_{p}\left(x\right)=S_{p}^{\prime }\left(x+v\left(x\right)\right).$$

This equation means there is timestamp consistency between the time surface $S_{p}$ and the shifted time surface $S_{p}^{\prime }$. Practically, the time surface is composed of a collection of events for a certain duration τ until a reference time $t_{0}$. Thus, the value range of the time surface is as follows:(5)$$S_{p}\left(x\right),S_{p}^{\prime }\left(x\right)\in \left[t_{0}-\tau ,t_{0}\right].$$

Equation (Equation 4) includes the non-linear term $S_{p}^{\prime }\left(x+v\left(x\right)\right)$ and it can be linearized around a close approximation $v^{0}$ to v using Taylor expansions as with the image-based TV-L1 optical flow estimation [26,27] as follows:(6)$$\rho_{p}\left(x,v\right)=\nabla S_{p}^{\prime }\left(x+v^{0}\left(x\right)\right)\cdot \left(v\left(x\right)-v^{0}\left(x\right)\right)+S_{p}^{\prime }\left(x+v^{0}\left(x\right)\right)-S_{p}\left(x\right)=0.$$
$\nabla S_{p}^{\prime }$ represents the spatial derivative of the shifted time surface.

The Surface Matching Loss is defined as summing for all pixels and each polarity after applying the L1 norm:(7)$$L_{\mathrm{surface}}\left(v\right)=\sum_{x}\sum_{p}{∥\rho_{p}\left(x,v\right)∥}_{1}.$$

In addition, since this loss is ill-posed alone, we add the following smoothness regularization term:(8)$$L_{\mathrm{smoothness}}\left(v\right)=\sum_{x}{∥\nabla v\left(x\right)∥}_{1}.$$

In general, optical flows are smoothly varying and have discontinuities at object edges. The smoothness regularization term aims to regularize the flow by reducing the difference between neighboring pixels. We chose the L1 norm, which constrains the optical flow smoothly while allowing for discontinuities in the optical flow. The L1 norm, which is generally considered to be robust to an outlier, is adopted to reduce the influence of the noise on the time surface. The estimated optical flow is obtained by minimizing this cost function as follows:(9)$$\min_{v}(L_{\mathrm{smoothness}}\left(v\right)+\lambda L_{\mathrm{surface}}\left(v\right)).$$

Note that the estimated optical flow v is at the 3-D point $\left(x,S_{p}\left(x\right)\right)$ on the time surface corresponding to the starting point of the arrow in Figure 2. That is, the reference time of the optical flow is not $\left(t_{0}-\tau \right)$, but the value of the time surface $S_{p}\left(x\right)$. Figure 3 shows a one-dimensional time surface for the varying motion speed for the duration of τ. As shown in the figure, the constant motion assumption in τ is not necessary because the optical flow is determined on the time surface. The assumption of constant speed is only necessary for the duration Δt to calculate the optical flow. It is easy to satisfy the assumption because Δt can be arbitrarily short, unlike image-based methods which have a fixed frame rate.

### 3.4. Comparison with Contrast Maximization

The most relevant to our work is the contrast maximization framework [10], which estimates the optical flow by searching for the trajectory that maximizes event alignment. In the contrast maximization, the IWE is created by warping events with the candidate velocity v and integrating them in an image as follows:(10)$$I\left(x;v\right)=\sum_{k=1}^{N_{e}}p_{k}\delta \left(x-x_{k}^{\prime }\left(v\right)\right),$$
where $x_{k}^{\prime }=x_{k}-\left(t_{k}-t_{\mathrm{ref}}\right)v$ is the warped event and $N_{e}$ is the number of the events. The variance of the IWE measures how well events agree with the candidate point trajectories as follows:(11)$$\mathrm{Var}\left(I\left(x;v\right)\right)=\frac{1}{N_{p}}\sum_{x}{\left(I\left(x;v\right)-\mu_{I}\right)}^{2},$$
where $\mu_{I}=\frac{1}{N_{p}}\sum_{x}I\left(x;v\right)$ is the mean of the IWE and $N_{p}$ is the number of pixels. By maximizing the variance of the IWE, the best trajectory can be obtained.

Figure 4 shows an overview of the contrast maximization approach [10] and the proposed surface-matching approach. Both of them measure the alignment of events across time. In the surface-matching approach, alignment of events is evaluated by the timestamp consistency referring to the fixed previous time surface *S* under the motion smoothness assumption in a rigid scene. On the other hand, there is no reference value in the variance loss, and alignment is implicitly measured by the contrast of the IWE generated from warped events.

## 4. Experiment

In order to verify the effectiveness of the proposed loss function, experiments about the loss landscape (Section 4.3) and optical flow estimation (Section 4.3) were performed.

### 4.1. Datasets

ESIMAn Open Event Camera Simulator (ESIM) [28] can accurately and efficiently simulate an event-based camera and output a set of events and the ground truth of optical flows with any camera motion and scene.MVSECThe Multi-Vehicle Stereo Event Camera Dataset (MVSEC) [29] contains an outdoor driving scene—by day and by night—and an indoor flight scene by the drone. The event-based camera used is mDAVIS-346B with a resolution of 346 × 260, which can capture general images simultaneously. The dataset provides the ground truth optical flow generated from depth maps by LiDAR and poses information by the Inertial Measurement Unit (IMU).HACDThe HVGA ATIS Corner Dataset (HACD) [25] is built with a recording of planar patterns to evaluate corner detectors. Those sequences were taken by an Asynchronous Time-based Image Sensor (ATIS) [30] with a resolution of 480 × 360. It also contains the position of markers at four corners of the poster, each 10 ms. With this information, the homography of the plane can be calculated, and the ground truth optical flow at any point on the poster can be obtained.

### 4.2. Loss Landscape

As in the experiment performed in [12], we investigated the shape of the loss function in the parameter space $v\in R^{2}$ when the optical flow is common at all pixels. In addition, the gradient for each pixel at a certain initial parameter was visualized. The gradients are simply compared, except for regularization.

Variance The variance loss represented by Equation (Equation 11). Events are warped by the optical flow $v\in R^{2}$ which is common at all pixels. The duration of the events used and the time interval of the optical flow were set to Δt+τ and Δt in order to match our method with the condition.Surface Matching Loss The proposed loss function represented by Equation (Equation 7). This loss is calculated by the difference between the time surface $S_{p}$ and the shifted time surface $S_{p}^{\prime }$ warped by the optical flow. The sign has been inverted to match the variance.

Event data is generated by ESIM in the translation on the image plane with various textures such as rectangle, checkerboard, brick and grass. The spatial resolution of the image and the patch size is set to 240×180 and 30.

Results and discussions

Consider the situation where a rectangle is translated on the image plane by a movement vector $v={\left(2,-1\right)}^{⊤}$(pix/Δt) as shown in Figure 5a. Figure 5b visualizes the landscape of the loss function in the magenta patch. Surface Matching Loss is wider in all directions, whereas the variance has a narrow peak.

Here, we compare the gradients of the two methods. The partial derivative of v with respect to the IWE is described as follows (refer to Equation (39) in [12] or the appendix of [19]):(12)$$\frac{\partial I}{\partial v}=-\sum_{k=1}^{N_{e}}p_{k}\nabla N\left(x-x_{k}^{\prime }\left(v\right);0,\epsilon^{2}\mathrm{Id}\right)\frac{\partial x_{k}^{\prime }\left(v\right)}{\partial v}$$
with $\partial x_{k}^{\prime }\left(v\right)/\partial v=-\left(t_{k}-t_{\mathrm{ref}}\right)$. When the motion model in the contrast maximization is 2-D optical flow, the motion model Jacobians $\partial x_{k}^{\prime }\left(v\right)/\partial v$ are equivalent to the timestamp. Equation (Equation 12) is the operation of the spatial derivative of the time surface after warping the events. Therefore, where the time surface is discontinuous, an erroneous gradient is calculated. As the parameter v approaches the true value, the gradient increase since the time surface of warped events becomes steep and the many events overlap.

On the other hand, the main gradient of our Surface Matching Loss is $\partial L_{\mathrm{surface}}/\partial v\propto \rho_{p}\nabla S_{p}^{\prime }$, which is the element product of the temporal difference and the spatial derivative of the time surface. In surface matching, as the parameter v approaches the true value, the gradient becomes gentler as the difference between the two surfaces becomes smaller.

The gradients in the patch are shown in Figure 5c,d with magenta arrows in the enlarged IWE and time surface. Figure 5c shows the situation when $v={\left(1,0\right)}^{⊤}$. It should be noted that the gradient in the *x* direction is further increased by integrating events in the places where many events overlap at the right side of the rectangle. This can make it difficult to reach a global optimal solution. In addition, it can be seen that the gradients of the variance loss are reversed at the right discontinuous edge formed by the new time stamp. On the other hand, in Surface Matching, the gradient is not reverse even at the edge of the surface and has the same length in the *x* and *y* direction.

Figure 5c shows the situation when $v={\left(2,0\right)}^{⊤}$, which is the optimal solution in *x* direction. However, there are largely reversed gradients on both sides of the event overlap. They cancel out on average if the image has two parameters, but can be unstable in pixel-wise estimation. On the other hand, Surface Matching Loss has no cost and no gradient at the place where it is properly matched, and the gradient in the *y* direction is relatively large.

In addition, the landscape of the loss function in scenes of checkerboard, brick, and grass are visualized in the Figure 6. In all scenes, variance has narrow peaks, while surface matching loss shows a gentle and stable landscape in all directions. In the grass scene, the landscape of Variance is almost elliptical, but in checkerboard and brick, the landscape is narrow along with the line texture.

### 4.3. Dense Optical Flow Estimation

We examine the effectiveness of the loss function by estimating dense optical flows. In this experiment, we focused on online optimization methods rather than learning-based methods, simply to compare loss functions excluding other factors. It should be noted that the proposed loss function can be used in combination with learning-based methods. The comparison was performed for the following optical flow estimation methods:Reconstruction The method for estimating optical flow simultaneously with luminance restoration [9]. A large number of optical flows and image parameters were optimized under a temporal smoothness assumption sequentially in the sliding window containing the events with a duration of 128 Δt.Variance The method of maximizing a variance of the IWE [10]. In [10], the optical flow parameters were common in the patches, but in order to make the conditions uniform, a dense optical flow estimation was performed by adding the L1 smoothness regularization as follows:
(13)$$\min_{v}\sum_{x}({∥\nabla v(x)∥}_{1}-\lambda \mathrm{Var}\left(I\left(x;v\left(x\right)\right)\right)).$$Events were warped by the optical flow at each pixel: $x_{k}^{\prime }=x_{k}-\left(t_{k}-t_{\mathrm{ref}}\right)v\left(x_{k}\right)$. The loss function is optimized by the primal-dual algorithm in the same way as the TV-L1 method [31]. The duration of the events used was set to Δt+τ to match our method.

Surface Matching Our proposed method rewritten as follows:

(14)$$\min_{v}\sum_{x}({∥\nabla v(x)∥}_{1}+\lambda \sum_{p}{∥\rho_{p}\left(x,v\right)∥}_{1}).$$

The parts of the time surface where no event occurred for τ were filled with 0. Since $S_{+}$ and $S_{-}$ have common optical flow parameters, optimization was performed for different polarities for simplicity. The loss function was optimized by the primal-dual algorithm in the same way as the images-based TV-L1 optical flow [27]. The given displacement field $v^{0}$ must be close to v, so that the approximation error of the Taylor expansions in Equation (Equation 6) is small. In practice, $v^{0}$ is given by the current estimate of v. For the first iteration, we set the initial value of $v^{0}$ to zero. The time surface $S_{p}$, $S_{p}^{\prime }\in \left[t_{0}-\tau ,t_{0}\right]$ was normalized to [0,255] and convolved with a Gaussian kernel of σ=0.8.

The Graphics Processing Unit (GPU) was used only as a method of Reconstruction. Reconstruction and Variance are our implementation, not the author implementation. The datasets used in this experiment are MVSEC and HACD composed of real data. The scene used are outdoor_day1, 2, outdoor_night1, 2, 3 and indoor_flying1, 2, 3 from MVSEC, and guernica_2, paris_2 and graffiti_2 from HACD. The duration of the optical flow Δt was set to 7.5 ms for outdoor scenes and 10 ms for indoor scenes from MVSEC, and to 5 ms for all scenes from HACD. The time width τ of the effective time surface was set to 10Δt in all scenes. In the experiments, Δt and τ were determined empirically for each scene. The regularization parameter λ was set to 0.15.

The average end-point error (AEE) was used as an evaluation metric:(15)$$AEE=\frac{1}{N}\sum_{x}{∥v_{\mathrm{est}}\left(x\right)-v_{\mathrm{gt}}\left(x\right)∥}_{2}.$$
$v_{\mathrm{est}}$ and $v_{\mathrm{gt}}$ indicate the estimated optical flow and the ground truth optical flow, respectively. The AEE was calculated only at the location where the event occurred. Furthermore, in MVSEC, the car reflection part at the bottom of the image was excluded and, in HACD, the outside of the poster was excluded. This is because they do not fit the model of generating the ground truth optical flow. *N* is the number of the valid pixels.

Qualitative result

The estimated optical flows in MVSEC and HACD are shown as Figure 7 and Figure 8, respectively. In the Variance method in all scenes, the gradient is estimated in different directions at some places because the time surface differentiates at discontinuous points.

Quantitative result

The result on the quantitative evaluation with the AEE is shown as Table 1. Our method is superior in two of the five sequences in MVSEC, all of the scenes in indoor_flying and HACD.

We consider the scene in which surface matching was inferior to reconstruction. Figure 9 shows the optical flow from the outdoor_day2 sequence. In the outdoor sequences of MVSEC, the optical flow is estimated up and down by the vibration of the car due to the unevenness of the road. The ground truth optical flow used the velocity data smoothed by the central moving average filter, so the effects of the vertical vibration were reduced [17]. Reconstruction used temporal smoothness regularization with a long time sliding window of 128Δt, which reduced the effects of vibration and increased the evaluation metric. Figure 10 shows the AEE and the pitch rate from IMU in the part of the outdoor_day2 scene. In the left part of the graph, where there is less vertical vibration on the ground, ours has a smaller error. The accuracy of our method can also be improved by adding temporal smoothness constraints. Conversely, the optical flow with Reconstruction is estimated to be small in HACD scenes where the direction of motion changes drastically.

The runtime was about 3 fps in a CPU-based MATLAB; however, the real-time implementation is possible because the optimization framework of the proposed method is the same as that of the image-based TV-L1 optical flow, which was implemented in real-time on the GPU [26]. Furthermore, in contrast to the image-based optical flow, which solves the problem coarse to fine by constructing a spatial pyramid due to the large frame interval, Δt can be arbitrarily small in the proposed method, eliminating the need for a spatial pyramid and enabling faster estimation.

### 4.4. Study on Hyperparameters

In our method, the effective time of the time surface τ and the duration of the optical flow Δt are important hyperparameters. It is preferable that Δt is as short as possible due to the optical flow constraint while the movement occurs. τ represents the time range of the time surface, and there is no constraint, however, if it is too large, the object will appear to have a long exposure time, and occlusion will occur. For this reason, we investigated the accuracy of optical flow estimation when Δt and τ were changed. The guernica scene was used for the evaluation.

The result of the study on hyperparameters is shown in Table 2. Looking at the τ=50ms column, the shorter Δt is, the higher the accuracy because the shorter time interval satisfies the constant motion conditions as hypothesized. In the Δt=5ms row, the shorter τ is, the higher the accuracy because the ground truth optical flow is at the reference time $\left(t_{0}-\tau \right)$. The best parameters were the smallest combination, (Δt,τ)=(2.5ms,25ms) in this experiment.

## 5. Conclusions

We propose a method of estimating the optical flow from event-based cameras by matching the time surface of the events. By optimizing the loss function that measures the timestamp consistency between the time surface and one that is slightly shifted, it is possible to estimate a dense optical flow from only the events without restoring the luminance. The experiments show that the proposed loss function has a stable landscape for the estimated parameter regardless of texture. It is shown that the gradient of our loss is equally calculated in all directions while that of the variance loss increases in the direction pushing the event to a line. Additionally, in the experiments involving dense optical flow estimation, our method added L1 smoothness regularization recoded with high accuracy.

Δt is an important parameter that must be adjusted for each scene. In future work, we would like to develop a mechanism to adaptively determine this parameter.

## Figures and Tables

**Figure 1 sensors-21-01150-f001:**
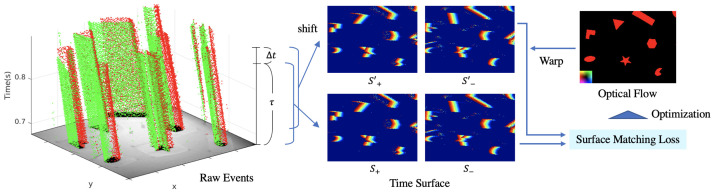
Overview of the proposed method. **Left**: The plot of the real event data [13] taken by DAVIS [14] in x−y−t space. The red and green dots indicate the positive and negative events respectively. **Middle**: Time surface and shifted time surface (**upper** and **bottom**) at each polarity (**left** and **right**). The event timestamp is color-coded (with red for the most recent and blue for the oldest part of the time surface). The brackets represent the time width in which the time surface is formed. **Right**: Warp the time surface by the optical flow parameters, measure the matching cost, and minimize it.

**Figure 2 sensors-21-01150-f002:**
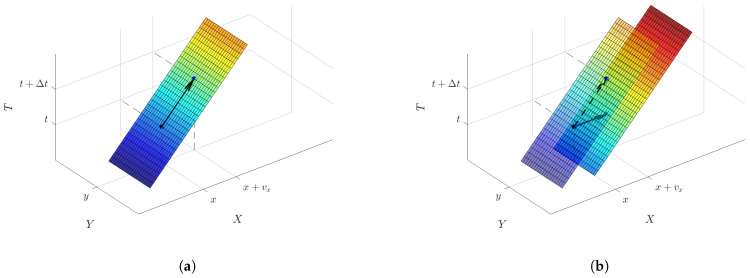
Principle of the time surface matching. (**a**) The time surface formed when a line segment over the Y direction moves in the X direction. The event timestamp is color-coded (with red for the most recent and blue for the oldest part of the time surface). The non-transparent surface in (**b**) is the Δt shifted time surface. Time Surface Matching Loss evaluated the consistency of the timestamps between the time surface and the shifted time surface.

**Figure 3 sensors-21-01150-f003:**
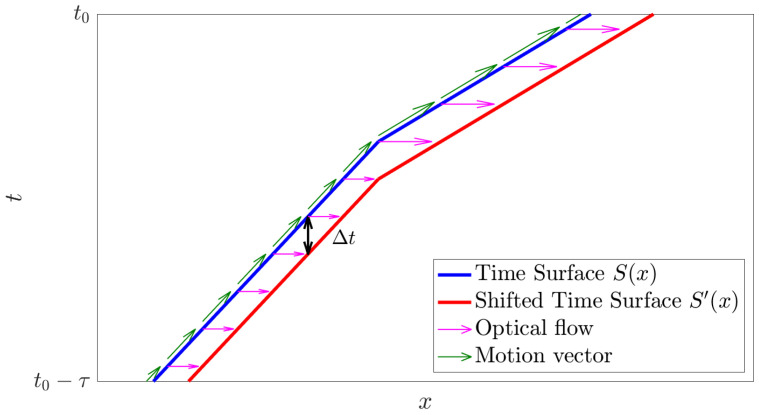
A one-dimensional time surface when the velocity changes over a time period of τ. Blue and red plots indicate the time surface and the shifted time surface, respectively. The green and magenta arrows indicate the motion vector in x−t space and the optical flow, respectively. The assumption of constant speed is only necessary for the very short interval Δt to calculate the optical flow.

**Figure 4 sensors-21-01150-f004:**
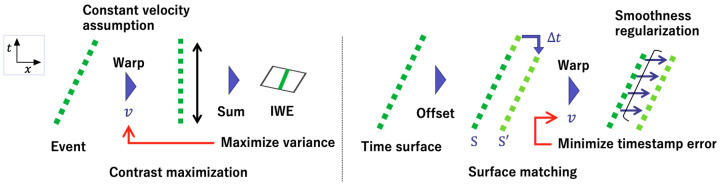
Comparison of the contrast maximization (**left**) and our surface matching approach (**right**). In the contrast maximization, alignment of events is implicitly measured by the contrast of the image of warped events (IWE) generated from warped events. In our surface-matching approach, alignment is evaluated by the timestamp consistency referring to the fixed previous time surface *S*. The green dots indicate the events in x−t space.

**Figure 5 sensors-21-01150-f005:**
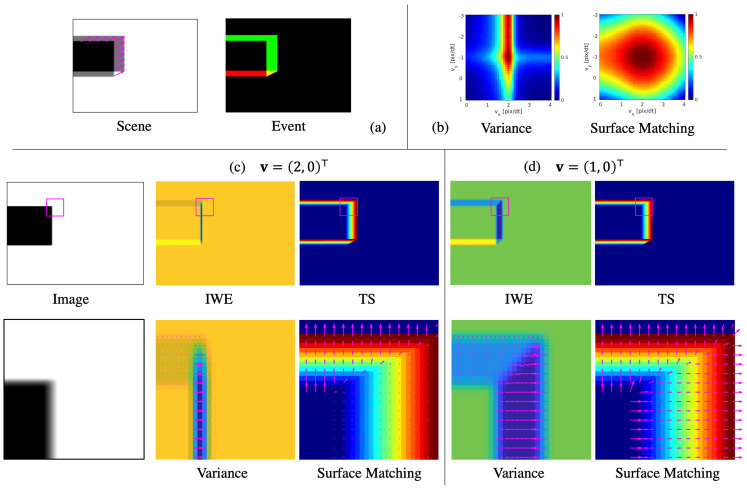
(**a**) Description of a rectangle scene and an event image where a rectangle is translating at $v={\left(2,-1\right)}^{⊤}$ on the image plane. (**b**) Plots of the losses as a function of the optical flow parameters $v\in R^{2}$. Each loss map is normalized by its maximum. Bottom of figure shows the brightness image, the IWE, time surface (TS) and those enlarged in a magenta patch when the loss functions are evaluated at (**c**) $v={\left(2,0\right)}^{⊤}$(pix/Δt) and (**d**) $v={\left(1,0\right)}^{⊤}$(pix/Δt). The magenta arrows show the gradient of variance for the IWE and Surface Matching Loss for the TS.

**Figure 6 sensors-21-01150-f006:**
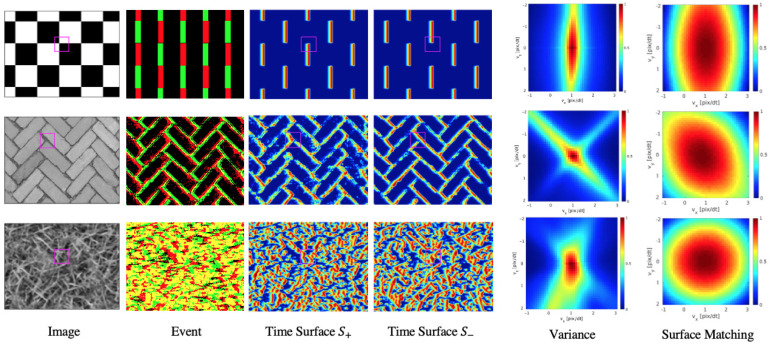
Visualization of loss landscape of Variance and Surface Matching Loss in the scene of checkerboard, brick and grass under the condition that the true optical flow is set to $v={\left(1,0\right)}^{⊤}$(pix/Δt). The loss functions are evaluated in a magenta patch.

**Figure 7 sensors-21-01150-f007:**
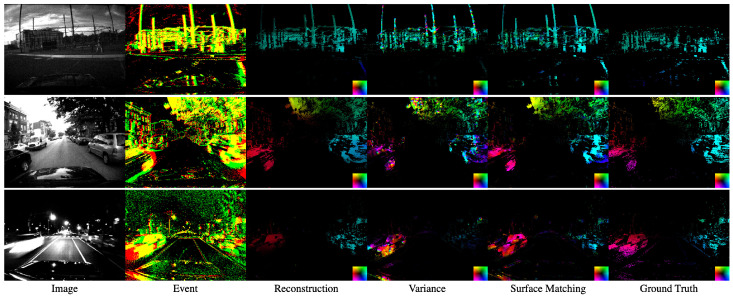
Qualitative results from Multi-Vehicle Stereo Event Camera Dataset (MVSEC). Left to right: the image, events, the optical flow by Reconstruction [9], Variance [10], Surface Matching (ours). Events are shown as accumulation for τ+Δt. The red and green events indicate each polarity and yellow shows that both of them occur at the pixel. Optical flow is colored by direction as the lower right colormap in each image.

**Figure 8 sensors-21-01150-f008:**
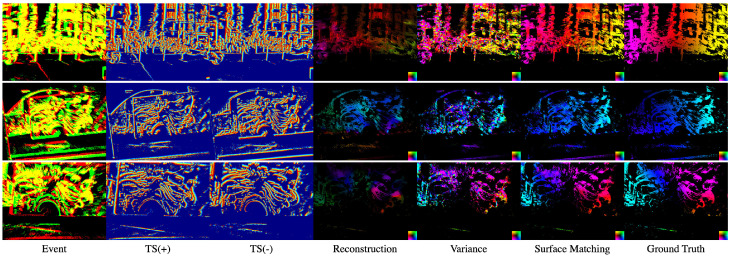
Qualitative results from HVGA Asynchronous Time-based Image Sensor (ATIS) Corner Dataset (HACD). Left to right: event data, positive time surface $S_{+}$, negative time surface $S_{-}$, the predicted optical flow by Reconstruction [9], Variance [10] and Surface Matching (ours). The red and green events indicate each polarity and yellow shows that both of them occur at the pixel. The time surfaces are colored with red for the most recent and blue for the oldest part of the time surface. Optical flow is colored by direction as the lower right colormap in each image.

**Figure 9 sensors-21-01150-f009:**
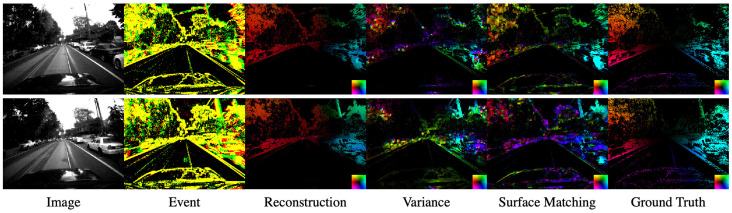
Qualitative results when shaking by the ground in an outdoor scene.

**Figure 10 sensors-21-01150-f010:**
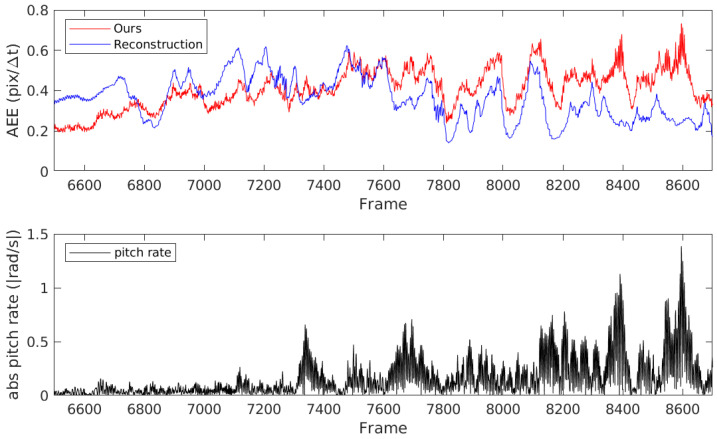
The average end-point error (**upper**) and the pitch rate from inertial measurement unit (**bottom**) in the part of the outdoor_day scene. In the left part of the graph, where there is less vertical vibration on the ground, ours has the smaller error.

**Table 1 sensors-21-01150-t001:** Quantitative results of optical flow estimation. The evaluation metric is the AEE (pix/Δt).

	Day 1	Day 2	Night 1	Night 2	Night 3	Flying 1	Flying 2	Flying 3	Guernica	Paris	Graffiti
Reconstruction [9]	0.267	0.307	0.283	0.313	0.365	0.348	0.525	0.468	1.99	2.79	1.91
Variance [10]	0.479	0.479	0.418	0.368	0.438	0.351	0.525	0.469	4.01	3.11	1.90
Surface Matching	0.257	0.350	0.334	0.363	0.356	0.278	0.422	0.377	1.50	2.30	1.36

**Table 2 sensors-21-01150-t002:** Results of study on hyperparameters, Δt and τ. The evaluation metric is average end-point error (AEE) (pix/Δt).

	τ	25 ms	50 ms	75 ms
Δt				
2.5 ms		0.95	1.14	-
5.0 ms		1.34	1.50	1.61
7.5 ms		-	1.74	1.81

## Data Availability

Data available in a publicly accessible repository. The data presented in this study are openly available in reference number [25,29].

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
