# Peer review of "Optical Flow Estimation by Matching Time Surface with Event-Based Cameras"

_sensors, 2021, doi:10.3390/s21041150_

Round 1
Reviewer 1 Report
This is an interesting, well written work. My main concern is its full understandability for readers whose everyday work is not mainly focused on event-based cameras. That is why I would suggest explaining several aspects of the presented research:
- What is v0 in (6)? How is it used in practice?
- In (6) function rho is equal to 0 and in (7) L1 norm of this value is calculated? What is optimized then (see next question). In this context L1 norm is simply an absolute value of rho?
- In (7) and (8) it is not fully clear if v is a vector (for one point) or vector field (for whole image) and if only one time point is considered or whole sequence of images (such sequences are analysed in tests). It would be good to specify what is the precise argument of L.
- Section 3.4 and figure 4 are not clear. I know that proper work is cited but it would be good for the paper if the contrast maximization method were described with more details. It would allow reader better understand figures 4, 5 and 6.
- In section 4.4 study of hyperparameters is presented. It should be, however, discussed earlier what is the influence of those parameters on results.
Author Response
Point 1: What is v0 in (6)? How is it used in practice?
Response 1: In practice, v0 is the current estimate of v.
For the first iteration, we set the initial value of v0 to 0.
We updated the manuscript by adding “The given displacement field v0 must be close to v, so that the approximation error of the Taylor expansions in eq. 6 is small.
In practice, v0 is given by the current estimate of v.
For the first iteration, we set the initial value of v0 to zero.” In L. 231 in Sec. 4.3.
Point 2: In (6) function rho is equal to 0 and in (7) L1 norm of this value is calculated? What is optimized then (see next question). In this context L1 norm is simply an absolute value of rho?
In (7) and (8) it is not fully clear if v is a vector (for one point) or vector field (for whole image) and if only one time point is considered or whole sequence of images (such sequences are analysed in tests). It would be good to specify what is the precise argument of L.
Response 2: rho represents the optical flow constraint. The equation is zero if the observation contains no noise and the displacement is optimal value. The displacement field is obtained by taking the absolute value of rho and optimizing it for v. As you pointed out, the arguments of L, rho, and v were not specified, which made it difficult to understand. In (7) and (8), v is a vector for one point x. We updated the manuscript by adding x to the argument of v and rho, and add v to the argument of L in Sec 3.3 and Sec 4.3.
Point 3: Section 3.4 and figure 4 are not clear. I know that proper work is cited but it would be good for the paper if the contrast maximization method were described with more details. It would allow reader better understand figures 4, 5 and 6.
Response 3: As you pointed out, the contrast maximization method should be described with more details. We updated the manuscript by moving the description about the objective function of the contrast maximization from Sec 4.2 to Sec.3.4. We add “The most relevant to our work is the contrast maximization framework [10], which estimates the optical flow by searching for the trajectory that maximizes event alignment. In the contrast maximization, the IWE is created by warping events with the candidate velocity v and integrating them in an image as follows:
Eq. 10
where is the warped event and Ne is the number of the events. The variance of the IWE measures how well events agree with the candidate point trajectories as follows:
Eq. 11
where is the mean of the IWE and Np is the number of pixels. By maximizing the variance of the IWE, the best trajectory can be obtained.” In L. 3.4. We also revised Figure 4 to include more information.
Point 4: In section 4.4 study of hyperparameters is presented. It should be, however, discussed earlier what is the influence of those parameters on results.
Response 4: In L.130 after eq.9 and in Sec.4.4, the parameters is mentioned, but there is not much explanation as you pointed out. We updated the manuscript by adding “τ represents the time range of the time surface, and there is no constraint, however, if it is too large, the object will appear to have a long exposure time, and occlusion will occur.”.
Reviewer 2 Report
The manuscript presented by Jun Nagata et al. entitled “Optical Flow Estimation by Matching Time Surface with Event-based Cameras” propose a new loss function measuring the timestamp consistency of the time surface for optical flow estimation using event-based cameras. By comparing the results with the traditional methods, authors show that the proposed loss function has a stable landscape for the estimated parameter regardless of texture. As authors said “\delta t is an important parameter…”, how to determine the most optimized parameter \ delta t? May I think \ delta t is hard to be optimized? Though, the authors present some interesting results by contrast with the previous works. Therefore, I suggest the manuscript can be accepted in current version.
Author Response
Point 1: As authors said “\delta t is an important parameter…”, how to determine the most optimized parameter \delta t? May I think \ delta t is hard to be optimized?
Response 1: Delta t should be small enough that the constraints of constant motion are satisfied, supporting the Taylor expansion in (6). Therefore, the parameter is expected to be scene-dependent, but by making it small enough, the effect can be reduced. This time, the parameters were determined empirically for each scene, but in the future, it will be necessary to adaptively determine the parameters from the scene as we stated in the section of conclusion. We updated the manuscript by adding “In the experiments, Δt and τ were determined empirically for each scene.” in L. 241 above eq. 14.
Reviewer 3 Report
The authors describe a new method for optical flow estimation using event-based cameras. Doing this, the authors exploit the asynchronous acquisition of information by this sensors, developing a loss function which measures time stamp.
I find this idea very clever, even it is specifically designed for event-based cameras and cannot be expanded to more commonly used cameras. By the way, for the specific subject of this paper, it falls within the scopes of the journal and it can be of interest for researchers dealing with event-based cameras.
The research design is appropriate, and compared with both SoA methods and SoA benchmark dataset. In the majority of the cases, the proposed methods yields better results.
I suggest some minor changes before it can be accepted.
1 - the english is good, but there are several glitches that can be easily removed with a more accurate proof reading
2 - Figure 8 is probably the most important and self-explanatory picture of the manuscript, I suggest to rework it to make it more visible for the reader.
3 - the state of the art section is a bit reductive for the subject, and several other outstanding papers should be cited (this is to expand the readability of the paper for non expert readers). In my opinion, it is too much focused only on flow estimation, but event-cameras deserves a wider explanation.
4 - table 1 summarizes the results, but I suggest the authors to add more details for those datasets and methods were their method does not reach better results (trying to motivate why)
5 - the discussion on hyper parameters left me a bit bewildered, as it sounds empirical and not motivated by a scientific reason. If in their tests Tao and Delta T have been chosen with some criteria, this should be clearly stated and linked with the future work section, I guess this will be the most important stuff to do for the future.
Author Response
Point 1: The English is good, but there are several glitches that can be easily removed with a more accurate proof reading
Response 1: Thank you for your kind advice. I would like to use MDPI's English editing service.
Point 2: Figure 8 is probably the most important and self-explanatory picture of the manuscript, I suggest to rework it to make it more visible for the reader.
Response 2: Figure 8 is one of the figures showing the results. As you pointed out, the figure was not fully explained. Therefore, we updated the manuscript by adding “The red and green events indicate each polarity and yellow shows that both of them occur at the pixel. The time surfaces are colored with red for the most recent and blue for oldest part of the time surface. Optical flow is colored by direction as the lower right colormap in each image.” In the description of Figure 8.
Point 3: The state of the art section is a bit reductive for the subject, and several other outstanding papers should be cited (this is to expand the readability of the paper for non expert readers). In my opinion, it is too much focused only on flow estimation, but event-cameras deserves a wider explanation.
Response 3: As you pointed out, several outstanding papers about other uses of the event cameras should be cited for those who are not familiar with the event cameras. We updated the manuscript by adding “, which make event cameras are more popular in many domains, such as mosaicing and tracking [2], 3D reconstruction [3], and high frame rate video generation [4].” in L. 15 in Sec. 1.
Point 4: Table 1 summarizes the results, but I suggest the authors to add more details for those datasets and methods were their method does not reach better results (trying to motivate why)
Response 4: As mentioned from L.255 to L.265, we attribute the worse performance than the previous work to the numbers of events used and the temporal smoothness term. These scenes contain a lot of ground vibrations. However, the ground truth optical flow was created with smooth camera motion with a central moving average filter applied [17]. we think the method of [9] is better in this situation because it uses more events (about 10 times our method) and adds temporal smoothness regularization with a long-time sliding window.
Point 5: The discussion on hyper parameters left me a bit bewildered, as it sounds empirical and not motivated by a scientific reason. If in their tests Tao and Delta T have been chosen with some criteria, this should be clearly stated and linked with the future work section, I guess this will be the most important stuff to do for the future.
Response 5: As you pointed out, the hyper parameters were determined empirically for each scene. The hyperparameters of time duration also exists in the comparison methods Reconstruction [9] and Variance [10]. This is an important issue in signal processing for event cameras. In the future, it will be necessary to adaptively determine the parameters from the scene.
We updated the manuscript by adding “In the experiments, Δt and τ were determined empirically for each scene.” in L. 241 above eq. 14.